# Long-Lived Individuals Show a Lower Burden of Variants Predisposing to Age-Related Diseases and a Higher Polygenic Longevity Score

**DOI:** 10.3390/ijms231810949

**Published:** 2022-09-19

**Authors:** Guillermo G. Torres, Janina Dose, Tim P. Hasenbein, Marianne Nygaard, Ben Krause-Kyora, Jonas Mengel-From, Kaare Christensen, Karen Andersen-Ranberg, Daniel Kolbe, Wolfgang Lieb, Matthias Laudes, Siegfried Görg, Stefan Schreiber, Andre Franke, Amke Caliebe, Gregor Kuhlenbäumer, Almut Nebel

**Affiliations:** 1Institute of Clinical Molecular Biology, Kiel University, University Hospital Schleswig-Holstein, Campus Kiel, Rosalind-Franklin-Str. 12, 24105 Kiel, Germany; 2Department of Neurology, Kiel University, University Hospital Schleswig-Holstein, Campus Kiel, Arnold-Heller-Str. 3, 24105 Kiel, Germany; 3Institute of Pharmacology and Toxicology, Technical University Munich, Biedersteiner Str. 29, 80802 Munich, Germany; 4Department of Public Health, Epidemiology, Biostatistics and Biodemography, University of Southern, Denmark, J.B. Winsloews Vej 9B, 5000 Odense, Denmark; 5Department of Clinical Genetics, Odense University Hospital, J.B. Winsloews Vej 4, 5000 Odense, Denmark; 6Department of Clinical Biochemistry, Odense University Hospital, Kløvervænget 47, 5000 Odense, Denmark; 7Department of Geriatric Medicine, Odense University Hospital, Kløvervænget 23, 5000 Odense, Denmark; 8Institute of Epidemiology and Biobank Popgen, Kiel University, University Hospital Schleswig-Holstein, Campus Kiel, Niemannsweg 11, 24105 Kiel, Germany; 9Clinic for Internal Medicine I, Division of Endocrinology, Diabetes and Clinical Nutrition, Kiel University, University Hospital Schleswig-Holstein, Campus Kiel, Arnold-Heller-Straße 3, 24105 Kiel, Germany; 10Institute of Transfusion Medicine, University Hospital Schleswig-Holstein, Campus Lübeck, Ratzeburger Allee 160, 23538 Lübeck, Germany; 11Institute of Medical Informatics and Statistics, Kiel University, University Hospital Schleswig-Holstein, Campus Kiel, Brunswiker Str. 10, 24105 Kiel, Germany

**Keywords:** longevity, PRS, healthy aging, age-related diseases

## Abstract

Longevity is a complex phenotype influenced by both environmental and genetic factors. The genetic contribution is estimated at about 25%. Despite extensive research efforts, only a few longevity genes have been validated across populations. Long-lived individuals (LLI) reach extreme ages with a relative low prevalence of chronic disability and major age-related diseases (ARDs). We tested whether the protection from ARDs in LLI can partly be attributed to genetic factors by calculating polygenic risk scores (PRSs) for seven common late-life diseases (Alzheimer’s disease (AD), atrial fibrillation (AF), coronary artery disease (CAD), colorectal cancer (CRC), ischemic stroke (ISS), Parkinson’s disease (PD) and type 2 diabetes (T2D)). The examined sample comprised 1351 German LLI (≥94 years, including 643 centenarians) and 4680 German younger controls. For all ARD-PRSs tested, the LLI had significantly lower scores than the younger control individuals (areas under the curve (AUCs): ISS = 0.59, *p* = 2.84 × 10^−35^; AD = 0.59, *p* = 3.16 × 10^−25^; AF = 0.57, *p* = 1.07 × 10^−16^; CAD = 0.56, *p* = 1.88 × 10^−12^; CRC = 0.52, *p* = 5.85 × 10^−3^; PD = 0.52, *p* = 1.91 × 10^−3^; T2D = 0.51, *p* = 2.61 × 10^−3^). We combined the individual ARD-PRSs into a meta-PRS (AUC = 0.64, *p* = 6.45 × 10^−15^). We also generated two genome-wide polygenic scores for longevity, one with and one without the *TOMM40*/*APOE*/*APOC1* gene region (AUC (incl. *TOMM40*/*APOE*/*APOC1*) = 0.56, *p* = 1.45 × 10^−5^, seven variants; AUC (excl. *TOMM40*/*APOE*/*APOC1*) = 0.55, *p* = 9.85 × 10^−3^, 10,361 variants). Furthermore, the inclusion of nine markers from the excluded region (not in LD with each other) plus the *APOE* haplotype into the model raised the AUC from 0.55 to 0.61. Thus, our results highlight the importance of *TOMM40*/*APOE*/*APOC1* as a longevity hub.

## 1. Introduction

Decades of extensive research on the etiology of human longevity have revealed the complex nature of this phenotype. In addition to healthy behavior, environment and chance, genetic factors have been shown to influence human longevity, with the genetic contribution estimated between 20–30% [1]. Despite an impressive variety of methodological approaches, the output of genetic longevity studies has been limited. Most of the longevity-associated variants exert only low to moderate effects. So far, just a relatively small number of genes (e.g., *APOE*, *FOXO3*, locus chr. 5q33.3, *CDKN2B*) have been confirmed to play a role in the phenotype across populations [2,3,4,5,6].

Long-lived individuals (LLI) reach extreme ages in relatively good health and are, therefore, considered models of healthy aging. Although LLI may suffer from comorbidities [7], they seem to age at a slower pace [4] and to avoid, postpone or survive age-related diseases (ARDs) [8]. It has not yet been conclusively clarified whether a lower genetic risk for ARDs contributes to longevity. Like longevity, most ARDs have a complex genetic architecture, but show a higher heritability than does longevity (e.g., up to 80% for type 2 diabetes (T2D) [9] and Alzheimer´s disease (AD) [10] and 40–60% for coronary artery disease (CAD) [11]). Studying genetic risk factors for major ARDs in LLI might help us identify variants relevant for both longevity and aging [12]. Taken further, a shared genetic architecture between longevity and ARDs might facilitate the prediction of an individual’s longevity potential. Data in the literature generally, although not consistently (e.g., [13]), strengthen a genetic link between longevity and ARDs [14,15,16,17]. For example, Fortney et al. successfully applied a disease-informed GWAS approach to detect new longevity genes [14]. They reported a large genetic overlap between longevity and AD and CAD, respectively, which is supported by other studies (e.g., [15,18,19,20]). For other ARDs, such as T2D, the evidence for a possible genetic link with human longevity is less clear (see, e.g., [14,15,20,21]).

The concept of “genome-wide polygenic score” (GPS) or “polygenic risk score” (PRS) has been successfully applied for several diseases, e.g., CAD, AD, and T2D [22]. GPSs for longevity have only been developed in a few studies so far [4,17,23]. Here, we applied published PRSs for seven common ARDs (AD, atrial fibrillation (AF), CAD, colorectal cancer (CRC), ischemic stroke (ISS), Parkinson’s disease (PD), and T2D) to a German longevity sample comprising 1351 LLI (≥94 years) including 643 centenarians, and 4680 younger controls (age range: 18–83 years, mean age: 50.5 years). We calculated the PRSs using genotyping data generated with the Illumina Infinium Global Screening Array-24 (GSAv1; 700,078 single-nucleotide variants (SNVs)). In addition, variants in and near the *APOE* gene have been repeatedly shown to have a strong negative impact on longevity [15] and thus could mask much smaller positive or negative effects of other SNVs. Therefore, we developed two new GPSs for longevity based on published summary statistics from the latest meta-GWAS on longevity [15], one with and one without considering SNVs in the *TOMM40*/*APOE*/*APOC1* gene region, and compared them with published longevity GPSs.

## 2. Results

### 2.1. Association Analyses Do Not Reveal New Longevity Loci

In the single-variant analyses, we considered either the whole data set or male/female and centenarian-only subsets, respectively (Appendix A). Most association signals disappeared after conditioning on variants in the *TOMM40/APOE/APOC1* region on chromosome 19, which is well-known for being negatively associated with human longevity [5,24]. Of note, also a signal in the gene *PVRL2* near *TOMM40/APOE/APOC1* did not survive our conditioning, supporting our previous observation that *PVRL2* most likely does not represent an independent genetic longevity-associated locus, but rather influences the phenotype via epigenetic mechanisms [25]. The remaining longevity-associated variants were either too rare to yield reliable results or they did not show an association in the Danish replication sample (Appendix A). In the gene-based analysis, only *TOMM40* remained significant after correction for multiple testing (*P*adj = 4.32 × 10^−2^; Appendix A). The poor outcome of the longevity association analyses highlights the need for multifactorial analyses and refined methodological strategies to unravel the etiology of human longevity.

### 2.2. ARD-PRS Distributions Show Significant Differences between LLI and Controls

We investigated the polygenic risk profile of LLI versus younger controls for seven common ARDs, namely CAD, AF, ISS, AD, T2D, CRC and PD, using published PRS models and the imputed and quality-controlled GSA dataset of our German sample. Fractions of 96.4% (AD), 93.2% (ISS), 89.3% (CAD), 88.7% (AF), 96.8% (CRC), 87% (T2D) and 95% (PD) of the published input-SNVs were covered by the imputed GSA data (Table 1). The calculated mean (or median) PRSs were always significantly lower for LLI compared with the younger controls (logistic model *p-*value < 0.05, Figure 1, Table 1). The potential of differentiating between LLI and controls, based on the single ARD-PRSs, was rather low, with the AUCs ranging between 0.51 and 0.59 (Table 1). The highest discrimination values were achieved by the AD-PRS and ISS-PRS (AUC: AD-PRS = 0.59, ISS-PRS = 0.58, Table 1). Sex and population substructure were found to be significant covariables in all PRS models (Appendix A), and their inclusion increased the discrimination between LLI and controls by 16% on average (Appendix A). This means that in our data, LLI and controls could be distinguished to a certain extent by sex and population stratification. Especially in terms of sex, this was to be expected due to the ratio of women to men of 2.7:1 in the sample of LLI (versus 0.7:1 in the controls). To improve the discrimination of LLI and controls based on ARD risk alleles, we used the single ARD-PRSs as influence variables and combined them into a meta-PRS. On the test dataset, this model exhibited an AUC of 0.64 (without accounting for sex and population substructure as covariables; Table 1) and yielded the coefficients shown in Appendix A. When we included sex and population substructure as covariables, the discriminatory capacity of the model increased by 10% (i.e., the AUC improved to 0.74; Appendix A). As expected, the inclusion of the covariables also increased the contribution of the single ARD-PRSs (Appendix A).

### 2.3. Longevity GPS Discriminates LLI and Controls with an AUC of 0.56

We calculated a GPS for longevity (GPSlong), i.e., “genome-wide”, for the German longevity sample using published summary statistics from a recent large longevity meta-GWAS [15]. GPSlong was based on 3,298,544 out of 8,597,396 published input-SNVs. The remaining 5,298,852 markers were not considered, as they were not present in our imputed and quality-controlled GSA dataset or because they had been removed due to ambiguity. We divided our study population into a training dataset of 3913 individuals (900 LLI; 3013 younger controls) and a test dataset of 1678 individuals (395 LLI; 1283 younger controls) (see Section 4). The GPSs had AUCs ranging from 0.56 to 0.58 (Figure 2a, Appendix A). The best polygenic score, termed GPSlong, in the discovery and test datasets exhibited an AUC of 0.58 and 0.56, respectively (*p*-value threshold for SNV selection = 5 × 10^−7^, McFadden R^2^ = 0.016 and = 0.098 in discovery and test datasets, respectively; Figure 2b, Appendix A). GPSlong used seven SNVs, four located in the *TOMM40*/*APOE*/*APOC1* region, two more in the vicinity of the genes *CEP89* (rs62127361, chromosome 19) and *GPR78* (rs7676745, chromosome 4), and one on chromosome 2 (rs116362179). LLI exhibited a significantly higher GPSlong than the younger controls (logistic model *p*-value = 1.45 × 10^−5^ in the test dataset).

We also calculated a second GPS (GPSlong II) after the removal of the *TOMM40*/*APOE*/*APOC1* region (126 variants were excluded from the genotyping data). The best performing score, termed GPSlong II, showed a significantly higher score for LLI than for younger controls, displaying an R^2^ of 0.09 and an AUC of 0.55 in the test dataset, i.e., it was equivalent to the AUC of GPSlong but was achieved with the inclusion of 10,361 SNVs (*p*-value threshold for SNV selection = 0.01; GPSlong II *p*-value = 9.85 × 10^-3^ (logistic model); Figure 2c,d, Appendix A). Next, we added all the SNVs from the *TOMM40/APOE/APOC1* region and the *APOE* haplotype to GPSlong II and applied stepwise backward regression. This last model (GPSlong II+) included GPS long II, 9 SNVs from the *TOMM40/APOE/APOC1* region (Appendix A) and the *APOE* haplotype. GPSlong II+ achieved an AUC of 0.61 (±0.032) in the test dataset.

Furthermore, we investigated the performance of recently published GPSs for longevity [23] and [17] in our cohort (Table 2). Specifically, we tested the best scores reported by Tesi et al. [23], “PRS-5” (best score excluding *APOE* variants) and “PRS-6” (best score including *APOE* variants) and the score published by Liu et al. [17], in the following designated as “PRS_Liu”. With regard to PRS-5 and PRS-6, our genotyping dataset covered 94.8% (91 SNVs) and 97% (324 SNVs) of the SNVs originally used in PRS-5 and PRS-6, respectively. Both genomic scores reached statistical significance in our logistic model and exhibited similar statistics and density distributions as reported in Tesi et al. [23] (Table 2). With respect to PRS_Liu, our dataset covered 86.1% (3414 SNVs) of the original input-SNVs. Although the score discriminated LLI and controls with *p* = 2.22 × 10^-3^, the AUC was only 0.53 in our data, and with that, considerably lower than the AUC of 0.77 reported by the authors (Table 2).

## 3. Discussion

Based on ARD-PRSs, the LLI in our study had a significantly lower genetic risk of developing ARDs than the individuals from the control sample. This was true for all seven ARDs analyzed, with the largest PRS effects for ISS and AD. We also showed that PRSs are more informative than single genetic variants, because PRSs capture most of the genetic variance due to common variants in a single number. Therefore, our results with respect to ARDs can end the long controversial debate on the contribution of genetic risk factors for these diseases to longevity.

The coverage of the input-SNVs from the initial ARD-PRSs in our data was good for all seven ARDs. The literature supports a link between disease-associated variants and human longevity, within and beyond the *TOMM40*/*APOE/APOC1* gene region [14,19,26,27]. Our results substantiated the relatively well-documented shared genetic component of longevity and predisposition to cardiovascular disease (CVD; significantly lower risk scores for CAD, AF and ISS in LLI compared to controls) as well as the genetic link with AD [14,15,18,19,20,28]. Additionally, we strengthened the evidence of a shared genetic architecture between longevity and T2D, as already indicated in, e.g., [15,20]. Moreover, the German LLI exhibited a lower PRS for CRC than the younger controls, supporting that the lower CRC prevalence, incidence and mortality in centenarians [29] has, at least in part, a genetic basis [30]. Remarkably, despite the relatively low incidence and heritability of PD [31,32] and probably even lower prevalence in centenarians [33], we detected minor differences in the joint impact of PD variants between LLI and younger controls.

Efforts in generating a proper GPS for longevity have been hampered mainly by (a) the generally small sample size of longevity-GWAS and the resulting inaccuracy of selected variants and effect sizes, (b) the non-standardized definition of cases and controls, (c) phenotype dilution (when parental longevity is used as phenotype) and (d) the lack of large datasets to validate and test the developed GPS. To date, only a handful of studies have developed GPSs for longevity to discriminate between LLI and controls [4,17,23]. In total, we constructed two new and replicated three published GPSs for longevity. Our first score, GPSlong, was based on the summary statistics from the last meta-GWAS on longevity [15]. The score included only seven input-SNVs, four of which were located in *TOMM40*/*APOE*/*APOC1*, and significantly differentiated LLI from younger controls with an AUC of 0.56. We achieved a similarly high AUC (AUC = 0.55) when we excluded the *TOMM40*/*APOE*/*APOC1* region (GPSlong II). Strikingly, in a model (GPSlong II+) including nine non-LD SNVs from the removed region, the *APOE* haplotype and the scores from GPSlong II, the AUC increased to 0.61. This finding confirms the very strong influence of *TOMM40/APOE/APOC1* in explaining longevity and its masking effect on other variants. The exclusion of this region allowed us to observe the cumulative effect of 10,361 SNVs with moderate to low effect size in GPSlong II. Interestingly, GPSlong II reached almost the same discriminative power as GPSlong. These analyses, in addition to showing significant differences in the distributions of LLI and controls, demonstrated a substantial genetic contribution to human longevity within, but also beyond, the well-described mortality-associated *TOMM40*/*APOE/APOC1* locus.

The third and fourth scores were both validations of scores published by Tesi et al. [23]. The variant coverage by our genotyping data was almost complete, and the scores provided statistical measures similar to those given in the original report. Notably, we validated these longevity scores despite different phenotype definitions (sporadic longevity versus parental longevity). The AUCs achieved in our cohort were 0.56 and 0.58, respectively. Unfortunately, no AUCs were reported by Tesi et al. [23]. The fifth score was a replication of a longevity-GPS recently published by Liu et al. [17] in a Chinese cohort. The authors had reported an extraordinarily high AUC of 0.77; however, although the score reached significance in our data, the AUC was as low as 0.53 despite relatively good input-SNV coverage (86.1%). This discrepancy might be partly due to population differences between European and Chinese individuals, even though Liu et al. [17] had used a European GWAS for PRS construction.

The development of the PRSs/GPSs carries both promises and limitations. They provide a quantitative measure of the predisposition of a phenotype based on a set of genetic variants. In particular, for longevity, a GPS could help researchers to improve the stratification of individuals into groups with significantly different odds. Although the currently existing, purely genetic scores do not individually predict whether a person will be long-lived or not (and this is also not to be expected given the relatively low heritability of human longevity), longevity scores could potentially be combined with, for instance, environmental, lifestyle or epigenetic factors to serve as a genetically informed phenotyping tool that may enhance the discrimination between LLI and controls in the future. Nonetheless, this type of predictive model is sensitive to the cryptic substructure of the population used (i.e., related to geography or participation bias [34]), and to confounding effects [35]. Moreover, the lack of standardized methods on how to integrate ancestry information [36], gene–environment interactions [37], and high-impact variants [38] and on how to find the best thresholds for SNV inclusion [35] and reliable metrics for selection of the best PRS [39] are factors influencing model performance and reproducibility.

## 4. Materials and Methods

Materials and methods are summarized in Figure 3.

### 4.1. Study Populations

The German longevity sample comprised 1351 unrelated LLI (mean age: 99 years, age range: 94–110 years), including 643 centenarians (≥100 years). The male:female ratio in the sample was 1:2.7. The participants were all of German ancestry and showed no overt signs of cognitive impairment. The recruitment of the German longevity sample was partly organized by the PopGen biobank and has been described in detail elsewhere [40]. Written informed consent to participate in the study was obtained from all participants, and the project was approved by the Ethics Committee of the Medical Faculty of Kiel University. The 4680 unrelated younger controls (age range: 18–83 years, mean age: 50.5 years) were recruited as part of the FoCus cohort [41] and as blood donors at the University Hospital Schleswig-Holstein in Kiel and Lübeck, Germany.

The 1003 Danish cases (mean age: 97.4 years, age range: 90.0–102.5 years, 75.7% women) were included in the present study for the validation of longevity association findings. The sample consisted of participants drawn from seven nation-wide surveys collected at the University of Southern Denmark: the Study of Danish Old Sibs (DOS), the 1905 Birth Cohort Study, the 1910 Birth Cohort Study, the 1911-12 Birth Cohort Study, the 1915 Birth Cohort Study, the Longitudinal Study of Danish Centenarians (LSDC), and the Longitudinal Study of Ageing Danish Twins (LSADT). Briefly, DOS was initiated in 2004 and included families in which at least two siblings were ≥90 years of age at intake. The 1905 Birth Cohort Study, 1910 Birth Cohort Study, 1915 Birth Cohort Study, and LSDC are prospective follow-up studies initiated in 1995, 1998, and 2010, when participants were 92–93, 95, and 100 years of age, respectively [42]. The 1911–1912 cohort study consisted of individuals who reached the age of 100 years in the period from May 2011 to July 2012 [43], and LSADT was initiated in 1995 and included Danish twins ≥70 years of age [44]. From DOS and LSADT, one individual from each sib-ship or twin pair was randomly selected among participants that had reached an age of at least 91 years for DOS, and 90 years for LSADT. From the 1905 and 1915 Birth Cohort Studies, participants were selected among individuals that had reached a minimum age of 96 years. The 738 controls (mean age: 66.3 years, age range: 55.9–79.9 years, 49.1% women) consisted of individuals recruited by the Danish Twin Registry (DTR) as part of the study of Middle-Aged Danish Twins (MADT). MADT was initiated in 1998 and included 4314 twins randomly chosen from each of the birth years 1931–1952 [44]. Surviving participants were revisited from 2008 to 2011, where the blood samples used for DNA extraction were collected. To ensure a control sample of unrelated individuals, only one twin from each twin pair was included. Written informed consents were obtained from all participants. Collection and use of biological material, and survey and registry information were approved by the Regional Committees on Health Research Ethics for Southern Denmark. The study was registered in SDU’s internal list (notification no. 11.163) and complies with the rules in the General Data Protection Regulation.

### 4.2. Variant Calling, Quality Control and Imputing for the German and Danish Study Populations

The German samples were genotyped on the Illumina Infinium Global Screening Array-24 (700,078 SNVs) (GSAv1, Illumina^®^ Inc., San Diego, CA, USA). Plink 1.9 [45] was used for per-individual and per-marker quality control (QC). In total, 431 individuals failed one or more of the following inclusion criteria: concordant sex information, missing genotype <8%, heterozygosity rate greater or lower than ±4 standard deviations from the mean, and no individual relatedness. Identity-by-descent (IBD) metric was used to estimate relatedness. In case of relatedness (IBD > 0.1875; halfway between the expected IBD for third- and second-degree relatives), only one individual was included in the analysis. Variants were excluded if the missing rate was higher than 5% and if they deviated from the Hardy–Weinberg equilibrium in control samples (HWE, *p* < 1 × 10^−5^). Following this QC procedure, 633,642 variants and 5600 individuals (1295 LLI and 4305 younger controls) remained for the analyses. Population stratification was evaluated with principal component analysis using a common set of independent markers (HapMap3 ancestry set from four ethnic populations). The principal components (PCs) were calculated with Plink 1.9 [45]. For further analyses, the first five PCs were used according to the scree plot. Outliers of the population substructure were identified based on these first five PCs and the local outlier factor metric (LOF > 2.1) [46]. Prior to imputation, a pre-imputation QC was implemented using the HRC-1000G-check-bim script v4.2.7 (https://www.well.ox.ac.uk/~wrayner/tools/#Checking accessed on 1 November 2020) to ensure good genotype estimations. Genotype imputation was performed using the secure cloud-based MIS [47] and selecting the Haplotype Reference Consortium HRC r1.1 2016 GRCh37/hg19 as a reference panel [48]. Phasing was performed by applying the Eagle 4.0 engine [48]. In a post-imputation QC, SNVs were excluded if they had an R2 < 0.75, deviated from the HWE (*p* < 1 × 10^−9^) in the control sample, had a genotype call rate <95% and/or showed an extremely low minor allele frequency (MAF < 1%). This resulted in 6,010,362 remaining autosomal variants.

The Danish cases were genotyped using the Illumina Human OmniExpress Array (Illumina^®^ Inc., San Diego, CA, USA) and imputed to the 1000 Genomes phase1 v3 reference panel using IMPUTE2 [49]. Pre-imputation quality control included filtering of SNPs on genotype call rate <95%, HWE *p* < 1 × 10^-4^, and MAF < 1%, and individuals on sample call rate <95%, relatedness and gender mismatch. Controls were genotyped using the Illumina Infinium PsychArray (Illumina^®^ Inc., San Diego, CA, USA) and imputed to the 1000 Genomes phase3 reference panel using IMPUTE2 [49]. Pre-imputation quality control included filtering SNPs on genotype call rate <98%, HWE *p* < 1 × 10^−6^, and MAF = 0, and individuals on sample call rate <99%, relatedness, and sex mismatch. After imputation, genotype probabilities were converted to hard-called genotypes in Plink using a cut-off of 90%, meaning that only genotypes with a probability of more than 90% were called. Variants with no genotype probabilities above 90% were set to missing.

### 4.3. Longevity Association Analyses in the German and Danish Study Populations

Single-variant association analysis was performed using the logistic regression test in Plink 1.9 [45] assuming an additive genetic model and adjusting for sex and the first five PCs for the German dataset. To test for independency of the candidate SNVs from the known longevity-associated locus *TOMM40*/*APOE*/*APOC1* [5,50], an additional logistic regression was employed, adjusting for the effects of the SNVs rs769449, rs157582 and rs150966173 in this region. Variants that reached a (borderline) significant *p*-value (*p* ≤ 0.05) after conditioning for the SNVs in the *APOE* gene were selected for replication in the Danish longevity sample. Gene-based association analysis was performed with both burden and non-burden approaches using the SKATO algorithm from the R-package SKAT [51]. False discovery rate [52] was used for multiple testing corrections.

### 4.4. Application of Published PRSs for Common ARDs in the German Longevity Study Population

PRSs describe the cumulative impact of many common variants on a specific disease. Variants are selected based on the association with the disease, taking into account the LD structure. The weights assigned to each genetic variant are usually derived from the effect sizes of the summary statistics from large GWAS. We investigated differences in the PRS distributions between LLI and younger controls for seven common ARDs, specifically CAD, AF, ISS, T2D, CRC, AD and PD. The diseases were selected because of their association with age [53] and the availability of summary statistics. In more detail, PRS summary statistics for CAD, AF, and T2D were acquired from [22], for ISS from [54], for AD from [55], for PD from [56], and for CRC from [57]. PRSs for ARDs were computed by first multiplying for each selected variant the genotype dose of the risk allele (coded by 0, 1 or 2) by its respective effect size (the log odds ratio (OR) from the GWAS summary statistics). Afterwards, the resulting values of all variants in the score were summed up using Plink 1.9 [45]. The areas under the curve (AUCs) of the ARD-PRSs, as a measure of the performance of the model, were calculated with pROC [58], and the significance of influence variables was assessed by logistic regression using the glmnet R package [59], once only for ARD-PRS as influence variable and once accounting additionally for sex and the first five ancestry PCs as covariables.

### 4.5. Computation of a Longevity-GPS

First, we defined a discovery dataset, sampling randomly 70% of our cohort (900 LLI and 3013 younger controls). The remaining 30% comprised the test dataset and were used for performance evaluation. For longevity, we computed a genome-wide polygenic score, GPSlong, based on the summary statistics of the latest meta-GWAS on longevity [15]. GPSlong was developed on the discovery dataset using PRSice-2 [60], including a linkage disequilibrium pruning approach by the clumping option (without reference panel, R2 and physical distance thresholds of 0.1 and 250 kb, respectively). In total, 11 different values for GPSlong were calculated using different *p*-value significance cut-offs for the SNVs, specifically *p* = 5 × 10^−8^*, p* = 5 × 10^−7^, *p* = 5 × 10^−6^, *p* = 5 × 10^−5^, *p* = 5 × 10^−4^, *p* = 3 × 10^−3^, *p* = 5 × 10^−3^, *p* = 1 × 10^−2^, *p* = 3 × 10^−2^, *p* = 5 × 10^−1^ and *p* = 1. The score with the best discriminative capacity (i.e., best at separating LLI and controls) was determined based on the maximal area under receiving-operating characteristic curve (AUC). The regression model was calculated using the glmnet R package [59], considering “being long-lived” as outcome and GPSlong as an influence variable with and without additional adjustment for sex and the first five ancestry PCs as predictors. AUC confidence intervals and McFadden’s pseudo-R^2^ were calculated with the pROC [58] and rcompanion [61] R packages, respectively. Due to the strong association of the *TOMM40*/*APOE*/*APOC1* region with longevity, an additional GPS was calculated after removal of the region (chr. 19: 45351000–45500000) (GPSlong II). To further assess the influence of this region, we employed a backward stepwise regression using GPSlong II scores, 30 SNVs (not in LD with any marker in the deleted region and MAF > 0.01) and *APOE* haplotype (*ε2*, *ε3*, and *ε4*; determined by the alleles at rs429358 and rs7412) as influential variables. The resulting model was denoted GPSlong II+. The performances of the models (i.e., GPSlong, GPSlong II and GPSlong II+) were evaluated in the test dataset using Plink 1.9. The comparison of the mean scores between LLI and younger controls and the replication of the longevity scores published by Tesi et al. [23] and Liu et al. [17] in our data were performed in the same way as described above for the ARD-PRSs.

### 4.6. MetaPRS Estimation

We applied an elastic-net logistic regression [62] to model the associations between longevity and the standardized ARD-PRSs into a metaPRS on the discovery dataset. This approach used the single ARD-PRSs as influence variables adjusting for sex and the first five ancestry PCs. The model was computed with the caret R package [63], model penalties were evaluated using 10-fold cross-validation, and class imbalance effects were controlled employing a smoothed bootstrap re-sampling approach [64]. The metaPRS was evaluated using the test dataset similar to GPS long and GPS long II.

## 5. Conclusions

Our PRSs analyses in the German sample showed an inverse correlation between longevity and the risk for the ARDs tested. The strong influence of the *TOMM40*/*APOE*/*APOC1* region on longevity was corroborated, supporting the view that this region is a longevity hub [25]. The PRS approach also helped us identify a considerable number of variants with mostly small effect sizes that influence longevity; however, future studies will be needed to refine PRS methodologies to better integrate genetic, environmental and disease-risk factors. Furthermore, exome-only GPSs could be very informative on the functional level. In general, an optimization of the existing models would require the implementation of additional validation cohorts.

## Figures and Tables

**Figure 1 ijms-23-10949-f001:**
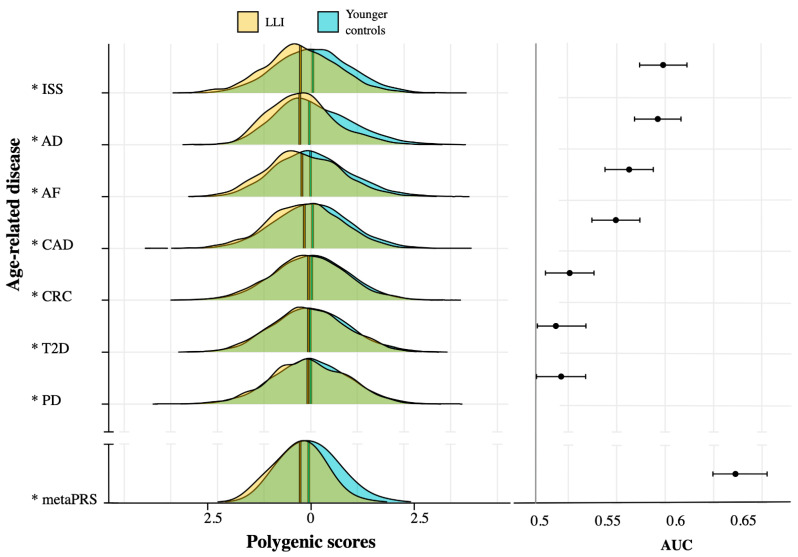
Scaled distribution of the polygenic risk scores of the single age-related diseases (ARD-PRSs) in long-lived individuals (LLI) and younger controls (Ctrl) and areas under the curve (AUCs) with 95% confidence intervals. Depicted values were taken from the model in which only the contributions of the PRSs were taken into account. Asterisks (*) indicate a significant relationship between the PRSs and longevity based on the logistic regression model. ISS, ischemic stroke; AD, Alzheimer’s disease; AF, atrial fibrillation; CAD, coronary artery disease; CRC, colorectal cancer; T2D, type 2 diabetes mellitus; PD, Parkinson’s disease; metaPRS, polygenic score calculated using the single ARD-PRS as influence variable. Within each distribution, the vertical line represents the mean. The black horizontal lines on the right represent the interquartile range and the circles represent the median AUC for each PRS.

**Figure 2 ijms-23-10949-f002:**
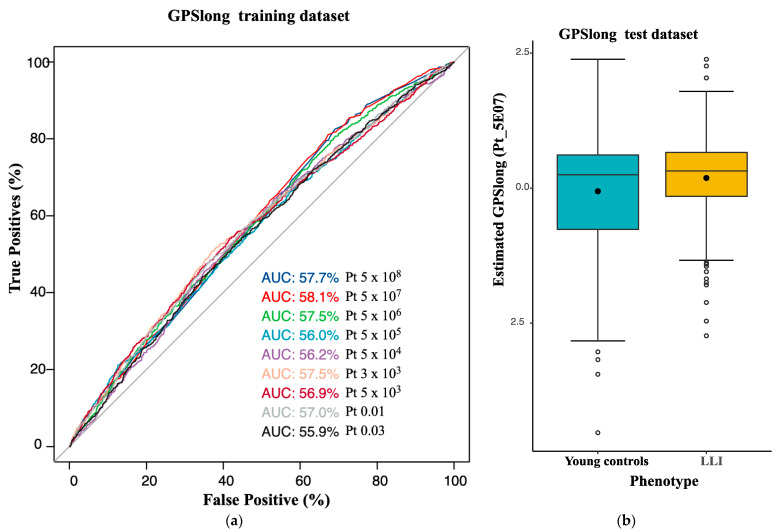
Longevity genome-wide polygenic scores (GPSlong and GPSlong II). (**a**,**c**) Discrimination potential of GPSlong (**a**) and GPSlong II (**c**) measured by the area under the curve (AUC) with different cut-offs. (**b**,**d**) Distribution of GPSlong (**b**) and GPSlong II (**d**) among the individuals. The values for LLI and younger controls in the test dataset are represented by boxplots. Pt denotes the different *p*-value significance cut-offs for the SNVs for GPS calculation.

**Figure 3 ijms-23-10949-f003:**
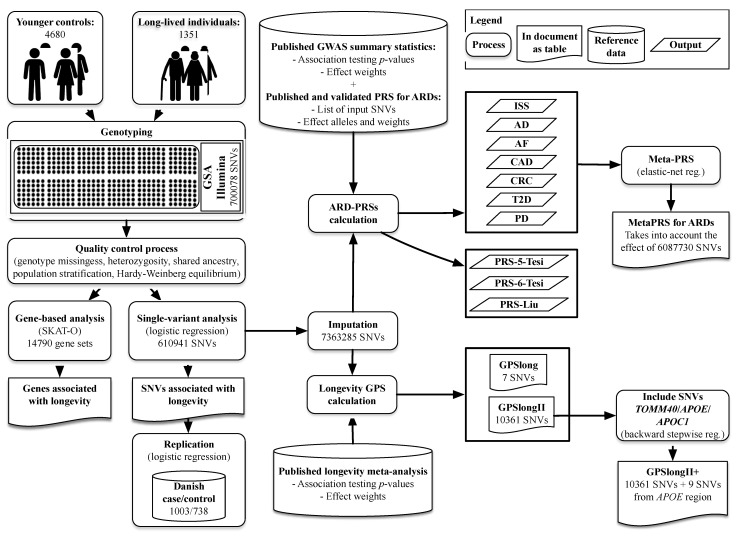
Study design and workflow of the longevity case–control GWAS, calculation of ARD-PRSs, meta-PRS for the diseases and the GPS for longevity. AD, Alzheimer´s disease; AF, atrial fibrillation; ARD, age-related disease; CAD, coronary artery disease; CRC, colorectal cancer; ISS, ischemic stroke; GPS, genome-wide polygenic score; GWAS, genome-wide association study; PD, Parkinson´s disease; PRS, polygenic risk score; SNV, single nucleotide variant; T2D, type 2 diabetes mellitus.

**Table 1 ijms-23-10949-t001:** Accuracy statistics for the calculation of each ARD-PRS in the German study population.

Age-Related Disease	AUC ^1^	AUC_L95	AUC_U95	Beta ^3^	OR ^4^	*p*-Value ^2^	PRS-Input-SNVs ^5^(No.)	Input-SNVs Covered ^6^(No. (%))
PD	0.52	0.50	0.53	−0.142	0.87	1.91 × 10^−3^	1805	1715 (95.01)
T2D	0.51	0.50	0.53	−0.056	0.95	2.61 × 10^−3^	6,917,436	6,024,432 (87.09)
CRC	0.52	0.50	0.54	−0.097	0.91	5.85 × 10^−3^	95	92 (96.84)
CAD	0.56	0.54	0.57	−0.159	0.85	1.88 × 10^−12^	6,630,150	5,920,526 (89.30)
AF	0.57	0.55	0.58	−0.170	0.84	1.07 × 10^−16^	6,730,541	5,973,364 (88.75)
ISS	0.59	0.57	0.61	−0.283	0.75	2.84 × 10^−35^	2,759,740	2,573,737 (93.26)
AD	0.59	0.57	0.60	−0.219	0.80	3.16 × 10^−25^	167	161 (96.41)
Meta-PRS ^7^	0.64	0.63	0.67	−0.403	0.67	6.45 × 10^−15^	6,087,730	6,087,730 (100)

AD, Alzheimer’s disease; AF, atrial fibrillation; ARD, age-related disease; AUC, area under the curve; CAD, coronary artery disease; CRC, colorectal cancer; ISS, ischemic stroke; PD, Parkinson’s disease; T2D, type 2 diabetes mellitus. ^1^ AUC, area under the curve; calculated for the logistic model using longevity (yes/no) as response variable and PRS only as influence variable; AUC_L95, AUC_U95, lower and upper confidence interval boundaries for AUC. ^2^ *p*-value of the logistic model for ARD-PRS. ^3^ Beta values (regression coefficients) for ARD-PRS. ^4^ OR, log(beta). ^5^ Input-SNVs from the original publications; for details see Section 4. ^6^ Covered by SNV genotyping/imputation in the German cohort. ^7^ Diagnostic measurements for single ARD-PRSs were calculated for the whole German dataset; the meta-PRS diagnostics were calculated for the test dataset.

**Table 2 ijms-23-10949-t002:** Statistics of GPSlong and GPSlong II in the German cohort (test dataset) as well as the results of the replication of the previously published scores compared to the reference data [17,23].

GPS	AUC ^1^	AUC_L95	AUC_U95	Beta ^2^	OR ^3^	*p*-Value ^4^	Number of SNVs ^5^
GPSlong (Germans)	0.56	0.53	0.58	0.281	1.32	1.45 × 10^−5^	7
GPSlongII (Germans)	0.55	0.52	0.55	0.158	1.17	9.85 × 10^−3^	10,361
PRS-5 (Germans)	0.56	0.54	0.57	0.223	1.25	5.33 × 10^−11^	324
PRS-5 [23]	NA	NA	NA	0.149	1.41	3.50 × 10^−9^	334
PRS-6 (Germans)	0.58	0.56	0.59	0.329	1.39	1.37 × 10^−20^	91
PRS-6 [23]	NA	0.57	0.61	0.158	1.44	7.30 × 10^−10^	96
PRS_Liu (Germans)	0.53	0.51	0.54	0.102	1.11	2.22 × 10^−3^	3414
PRS_Liu [17]	0.76					1.90 × 10^−5^	3966

AUC, area under the curve; GPS, genome-wide polygenic score; OR, odds ratio; PRS, polygenic risk score; SNV, single-nucleotide variant. ^1^ Calculated for the logistic model using longevity (yes/no) as response variable and GPS/PRS, sex and five principal components (population substructure) as influence variables; AUC_L95, AUC_U95, lower and upper confidence interval boundaries for AUC. ^2^ Beta values (regression coefficients) for GPS. ^3^ OR, log(beta). ^4^ *p*-value of the logistic model for GPS/PRS. ^5^ The number of variants that contributed to each GPS/PRS.

## Data Availability

All German samples and information on their corresponding phenotypes were obtained from the PopGen Biobank (Schleswig-Holstein, Germany) and can be accessed through a Material Data Access Form. Information about the Material Data Access Form and how to apply can be found at https://www.uksh.de/p2n/Information+for+Researchers.html.

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
