# Peer review of "Long-Lived Individuals Show a Lower Burden of Variants Predisposing to Age-Related Diseases and a Higher Polygenic Longevity Score"

_ijms, 2022, doi:10.3390/ijms231810949_

Round 1

Reviewer 1 Report

Gist/summary:  The authors come up with an interesting work on assessing long-lived individuals (LLI) and their predisposition to age-related diseases ARD) and a higher polygenic longevity score (PLS).  The authors come up with a chohort of 5000+ individuals and srceen 60k+ variants with detailed statistics and p-value correlations, furthe remploying the area under curve (AUC) and linear regression statistics.

Limitations: The manuscript is well written accomplishing major variants associated with LLI in the cohort.  While the authors employ Infinium array, the chance of identifying a ncRNA SNP would be nullified even otherwise exome sequencing of longeivity cohort was not used in their analyses. However,  the results they establish on such a large cohort with discrete statistics exemplified and corroborates the methods and they are in agreement. 

In the setting of introduction, there must be a rationale on APOE/TOMM genes and the LD associated with it for naive audience. 

The authors' rationale of classifying the variant as a risk allele, for example rs7153036 with MAF 0.245 as averse to 0.275: https://www.ncbi.nlm.nih.gov/snp/?term=rs7153036 in Genome_DK is with differenta ssociation values.  Why and how this MAF could be significant just because it is associated with  significant enriched p-value may not augur well  (significant findings as the authors claim).  Please justify 

Could the authors get a chance to look into this cohort exploited by Nygaard et al, a Canadian group? https://www.ncbi.nlm.nih.gov/pmc/articles/PMC6696723/  The chance of PRS/GRS with genotyping/Infinium array tends to be lower than the exome.  It would have been nice ha dthe uathors used this cohort for comparison so as to check the genetic variation.

Third, genetic variation is best seen in ncRNAs and in any case,   could the authors check the SNVs in and outside infinium array-24?  These could be wonderful leads

A pictorial methodology would be very nice to capture the wonderful materials an ddetailed statistics the authors have delved upon  

Minor but essential:

L266: Pl use plural:  scoreS

L396:  correctionS

Reviewer 2 Report

Although optional, ending with a Conclusions section is more clarifying.
